**DOI: 10.1038/ncomms13914**　　**OPEN**

# Nonlinear detection of secondary isotopic chemical shifts in NMR through spin noise

Maria Theresia Pöschko[1], Victor V. Rodin[1], Judith Schlagnitweit[1,†], Norbert Müller[1] & Hervé Desvaux[2]

The detection of minor species in the presence of large amounts of similar main components remains a key challenge in analytical chemistry, for instance, to obtain isotopic fingerprints. As an alternative to the classical NMR scheme based on coherent excitation and detection, here we introduce an approach based on spin-noise detection. Chemical shifts and transverse relaxation rates are determined using only the detection circuit. Thanks to a nonlinear effect in mixtures with small chemical shift dispersion, small signals on top of a larger one can be observed with increased sensitivity as bumps on a dip; the latter being the signature of the main magnetization. Experimental observations are underpinned by an analytical theory: the coupling between the magnetization and the coil provides an amplified detection capability of both small static magnetic field inhomogeneities and small NMR signals. This is illustrated by two-bond $^{12}C/^{13}C$ isotopic measurements.

[1] Institute of Organic Chemistry, Johannes Kepler University Linz, Altenbergerstraße 69, 4040 Linz, Austria. [2] NIMBE, CEA, CNRS, Université Paris-Saclay, CEA/Saclay, 91191 Gif-sur-Yvette, France. † Present address: Departement of Medical Biochemistry and Biophysics, Karolinska Institute, 171 77 Stockholm, Sweden. Correspondence and requests for materials should be addressed to H.D. (email: herve.desvaux@cea.fr).

A major challenge in analytical chemistry is the detection of small amounts in the presence of an excess of a spectroscopically similar substance. It occurs, for instance, when low-abundance isotopic species are exploited for obtaining a specific 'fingerprint', providing attractive applications, for instance, for characterizing the origin of natural products[1], for environmental studies[2], for toxicology[3] or for proteomics[4]. When this challenge is addressed by nuclear magnetic resonance (NMR) three issues appear for distinguishing minor components from the major one: the spectral resolution, the dynamic range and potentially a nonlinear effect, called radiation damping[5,6], which results from the feed-back field induced by the precessing magnetization of the major component. It is particularly manifest when state-of-the-art high sensitivity receiving circuits are used. Radiation damping tends to broaden and shift the main resonances, totally obscuring small signals in the vicinity of large ones. To circumvent these effects, one remedy resides in diluting the mixture until the main compound does not cause these effects anymore. However, this also dilutes the minor species and therefore leads to low signal-to-noise ratios, long measurement times and high analysis costs, since NMR *per se* is not very sensitive. Moreover, changing the concentration causes a variation of the chemical shifts and therefore may interfere with the final aim of the analysis. These issues are encountered in the specific case of NMR isotopic measurements in $^1H$ NMR spectra due to low-natural abundance isotopes such as $^{13}C$. Isotope effects on chemical shifts (IECS)[7] induce small differences in chemical shift (in the p.p.b. range), requiring special care for their measurements, if radiation damping affects the main isotopomer component. While detection of one-bond $^{13}C-^1H$ isotopic effects on high-resolution liquid-state NMR spectrometers is facilitated through the appearance of so-called satellite signals split due to the large $^1J$ scalar coupling, the small $^1H$ signals caused by two-bond isotopic effects are not easily observable experimentally. Nevertheless long-range IECS have been detected either directly through ultra-high-resolution spectroscopy[8] or indirectly through two-dimensional NMR experiments[9].

The fluctuation-dissipation theorem in its acceptation of fluctuation-response[10,11] provides an alternative to the excitation-response detection scheme: monitoring fluctuations from which correlation functions and spectral densities are computed, gives access to susceptibilities. As a consequence, instead of obtaining the NMR spectra by the classical means–radio-frequency (rf) excitation of the nuclear magnetization, followed by the measurement of the induction created by the precessing magnetization which relaxes towards thermal equilibrium–, it is possible to obtain pieces of information on the nuclear susceptibility, that is, chemical shift and transverse relaxation rate, by only considering fluctuations at probe outputs. This last scheme requires a very simplified electronic apparatus, since only the detection circuit, and not the rf excitation one with amplifiers, synthesizers, transmitter–receiver–switch,… is needed. This detection principle is known in NMR as nuclear spin noise[12–14]. It has been used previously for optimizing NMR probe sensitivity[15–18] and to achieve NMR detection without rf excitation[19,20] which is useful under high polarization conditions[21–23]. Here, we show that minor components, such as isotopic satellites, in concentrated solution or neat liquids can be more easily and more reliably determined by spin-noise detection than by the usual methods involving rf excitation, in particular when the electronic detection circuit is at lower temperature than the sample.

We derive the theoretical analytical expression of nuclear spin-noise spectra in the presence of more than one spin species coupled to a resonant rf detection circuit. Three noise-source contributions affecting the spin dynamics are considered: (i) the

fluctuations of the transverse magnetization, as predicted by Bloch[24], (ii) the Nyquist noise due to the coil resistance, (these two sources are those usually considered[12–14]) (iii) and the Nyquist noise due to the finite impedance of the preamplifier, as recently introduced[25]. These different fluctuations act on the nuclear magnetization components. Moreover nonlinear contributions resulting from the feed-back fields induced by electrical current flowing inside the rf coil due to the precessing magnetization superimpose. The generality of the theoretical treatment allows us to address also static magnetic field inhomogeneities. We show that a small magnetic field gradient induces the appearance of a narrow spectral feature in the broad resonance line shape centered at the exact chemical shift. It is indeed not affected by the frequency pushing contribution due to the detection coil[5,6,14,18,26,27]. Thus it provides a tuning independent chemical shift reference. To illustrate potential applications, we determine the two-bond $^{13}C/^{12}C$ IECS, denoted $^2\Delta\ ^1H(C)$, directly from $^1H$ nuclear spin-noise spectra. The reliability and efficiency of this detection mode is owed to (i) the nonlinear enhancement due to the coupling between the nuclear spins and the detection coil, (ii) the strong dependence of this amplification effect on the total longitudinal magnetization (as a combined result of both concentration and polarization), when one employs a cooled-coil probe and (iii) the independence of any rf pulse calibration. The enhanced detection capability opens the route for improved electronic circuit designs useful for classical NMR but also specific for *in situ* passive observation based on spin noise with a very simplified electronic equipment, that is, a simple spectrum analyzer.

## Results

**Analytical expression for the spin-noise spectral density.** Until recently, the theoretical description of NMR spin noise mainly derived from the study of McCoy and Ernst[13], which was later refined to provide expressions for cooled-coil probes[16], or for hyperpolarized systems[21]. To analyse isotopic effects, the theoretical approach needs to be extended since the spin ensemble comprises more than one species, and their associated magnetizations can interact through cross-precession mediated by the radiation damping feed-back field[28]. In other words, the strong transverse magnetization $\mathscr{M}_r^a$ of spin species $a$, precessing at Larmor frequency $\omega^a$ induces a current in the resonant rf-coil. This current creates an rf feed-back field whose frequency is $\omega^a$ at first level of approximation, and which acts on the magnetization $\mathscr{M}^a$ as well as those of the other spin species, $\mathscr{M}^{m\neq a}$, in an non-negligible way mainly when their Larmor frequencies, $\omega^m$, are close to $\omega^a$. This ultimately leads to the appearance of unexpected resonance line shapes[28]. To describe this interference effect, an extension of the McCoy and Ernst derivation[13], which is based on electrical circuit description with a single nuclear susceptibility affecting the coil inductance, is not straightforward. Instead, in this regime where the contribution from spontaneous emission coupled to the cavity is weak[29], we use generalized Bloch equations in which the rf fields include feed-back fields and randomly fluctuating fields[12], for which several noise sources were considered: those resulting from the transverse magnetization and those resulting from the electronic Nyquist noise due to the resistive elements of the circuit.

In noise spectroscopy, the relevant spectroscopic information is deduced from spectral densities, which are computed as a power spectrum. Assuming that the electronic detection circuit can be represented by an RLC equivalent circuit[26] (coil of impedance $L$ and resistance $R$ in parallel to the capacitance $C$), the general equation for the noise voltage spectral density $W^U(\omega)$, valid whatever the number of spins $n$, their polarization and the

relative temperatures of the coil and the spins, can be derived at any electronic circuit resonance frequency (see 'Methods section' and Supplementary Methods):

$$W^{U} = \mathscr{A} \frac{1 + 2\theta \sum_m \chi_{eq}^{m\prime\prime}(\delta\omega^m)|k(\Delta\omega_{LC})|^2/k\prime(\Delta\omega_{LC})}{\left|1 + 2ik(\Delta\omega_{LC})\sum_m K^m\chi(\delta\omega^m)\right|^2} + W_a^U \quad (1)$$

where the spectrally relevant pieces of information are encoded in the nuclear susceptibility $\chi^m$ of the $m$th spin species:

$$\chi^m = \chi^{m\prime} - i\chi^{m\prime\prime} = \frac{\mu_0\gamma\mathscr{M}_z^m}{2}\frac{1}{-i\delta\omega^m + \lambda_2^m} \quad (2)$$

through its transverse relaxation rate $\lambda_2^m$ and Larmor frequency $\omega^m$ with $\delta\omega^m = \omega - \omega^m$. In equation (1), $\mathscr{A}$ is a constant dependent on the electronic circuit components, $K^m$ is the enhancement factor of nuclear polarization relative to thermal equilibrium, the symbol $\theta$ indicates the sample-to-coil temperature ratio. $k(\Delta\omega_{LC})$ is the complex number coefficient of proportionality between the rf feed-back field and the precessing magnetization. It is dependent on the tuning mismatch $\Delta\omega_{LC} = \omega_0 - \omega_{LC}$ with $\omega_{LC} = 1/\sqrt{LC_T}$ with $Q = L\omega_0/R$, the coil (of impedance $L$ and resistance $R$) quality factor, and $C_T$, the capacitance in parallel to the coil at the ends of which the voltage is measured[26]. $W_a^U$ is a constant noise contribution due to the measurement circuit. For a circuit whose electronic resonance $\omega_{LC}$ is exactly equal to the average spin Larmor frequency $\omega_0$, the noise voltage spectral density at frequency $\omega$ simplifies:

$$W^U(\omega) = \frac{2Qk_BT}{\pi C_T\omega_0}\frac{1 + \theta\sum_m \frac{\lambda_{r0}^m\lambda_2^m}{\lambda_2^{m2}+\delta\omega^{m2}}}{\left(1 + \sum_m \frac{\lambda_2^m\lambda_r^m}{\lambda_2^{m2}+\delta\omega^{m2}}\right)^2 + \left(\sum_m \frac{\lambda_r^m\delta\omega^m}{\lambda_2^{m2}+\delta\omega^{m2}}\right)^2} + W_a^U \quad (3)$$

Finally, the radiation damping rate $\lambda_r^m$ is:

$$\lambda_r^m = \frac{\mu_0}{2}\eta Q\gamma\mathscr{M}_z^m = \frac{\mu_0}{2}\eta Q\gamma K^m c^m\mathscr{M}^0 \quad (4)$$

with $\eta$ the coil filling factor and $\gamma$ the gyromagnetic ratio. The longitudinal magnetization of species $m$, $\mathscr{M}_z^m$, can be related to a reference magnetization $\mathscr{M}^0$ by introducing the ratio of concentrations $c^m$. When the magnetization is at thermal equilibrium ($K^m = 1$) this radiation damping rate is denoted $\lambda_{r0}^m$.

The RLC electronic circuit model was, however, recently shown to be insufficient for representing all experimental aspects in particular for cooled-coil probes[18]. At the spin-noise tuning optimum (SNTO), where purely absorptive Lorentzian shape spin-noise spectra are observed[16,25], the imaginary part of the radiation damping effect does not necessarily vanish, which causes frequency pushing of the resonance signal proportional to its associated radiation damping rate[14,26,27]. To resolve this contradiction between theory and experiment, it is necessary to consider the entire detection circuit and, in particular, the potential dephasing effects owed to the transmission line and the fluctuating rf fields induced by the electronic Nyquist noise resulting from a finite preamplifier resistance[25]. The detailed analytical extension to the present multi-spin system is inhibited by the large number of unknown electronic parameters of real cooled-coil probes. Nevertheless, a formal demonstration using a transmission matrix $T$ describing the voltage and intensity relations at the preamplifier input impedance and at the coil entries allows us to obtain a general equation for the noise voltage spectral density, valid for any tuning condition and an arbitrary number of spin species:

$$W = \mathscr{A}\frac{1 + 2\theta\sum_m \chi_{eq}^{m\prime\prime}(\delta\omega^m)|k(\Delta\omega_{LC})|^2/k\prime(\Delta\omega_{LC})}{\left|1 + 2ik(\Delta\omega_{LC})\sum_m K^m\chi(\delta\omega^m)\right|^2}$$
$$+ \mathscr{B}\frac{\left|1 + 2i\zeta\sum_m K^m\chi(\delta\omega^m)\right|^2}{\left|1 + 2ik(\Delta\omega_{LC})\sum_m K^m\chi(\delta\omega^m)\right|^2} + W_a^U \quad (5)$$

where $\mathscr{A}$ are $\mathscr{B}$ two real positive coefficients, and $\zeta$ is a complex number. These three parameters depend on the electronic components, and in particular, for $\zeta$, on the length of transmission line between the preamplifier and the coil.

Analysis of equation (5) corroborates the conclusions obtained for a single-spin species that the spin-noise spectral density can be described as the real part of a generalized Lorentzian function, that is, a mixture of absorptive and dispersive Lorentzian functions[25]. But if the values of the electronic components are unknown, a number of solutions exists for the combination of two Lorentzian functions. As a consequence, the best-fit theoretical curve of equation (5) to the experimental nuclear spin-noise spectra exhibiting a single line does not provide a unique set of parameters.

**Effects of weak gradients on spin-noise spectra.** When acquiring spin-noise spectra in the presence of weak gradients, that is, gradients which cause a line broadening equal to or less than the radiation damping rate, a complex line shape is observed for the radiation-damped signals. In Fig. 1, we compare the experimental line shapes of spin-noise spectra of a sample of 90% $H_2O$:10% $D_2O$ to simulations obtained by a programme based on equation (5), which simulates a magnetic field gradient along the $z$ axis.

To achieve this, the simulated sample is divided into a number of equal slices with linearly increasing magnetic field strengths. For each slice the magnetization induces a feed-back field and each slice experiences the sum of all feed-back fields corresponding to all slices. As a consequence, the cross-precession terms between the different magnetization volumes are taken into account. The line shapes obtained by these simulations agree reasonably well with the experimentally observed ones, considering the non-ideal situation of sample boundaries and gradient linearity in the cryogenically cooled probe used. Reproduction of these experimental gradient interference effects provides a confirmation for the suitability of the model. One remarkable feature of this interference effect between static field gradients and radiation damping as evident from Fig. 1 is the fact that the positive peak imaging the gradient profile (that is, a one-dimensional image of the sample along the $z$ axis) is always centered at the Larmor frequency provided the applied gradient is symmetrical with respect to the origin ($z = 0$) at the centre of the sample. There is no displacement of this one-dimensional spin-noise image[19] by the frequency pushing effect caused by radiation damping, whatever the probe's tuning (SNTO, FSTO, frequency shift tuning optimum, that is, the tuning condition at which the frequency pushing contribution vanishes, or conventional tuning optimum[15,16,18]). The centre of the gradient profile can thus be used to determine the chemical shift, irrespective of probe tuning, which by itself has a significant influence on the observed peak positions in spin-noise spectra[16,18] and small-flip-angle spectra[28]. This interference of weak gradients with radiation damping also gives rise to very small positive deviations from the smooth line shape of radiation-damped signals in spin-noise spectra as illustrated in the Supplementary Fig. 1. They are an indication

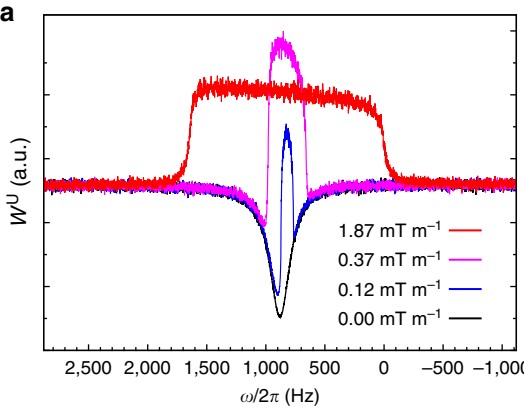

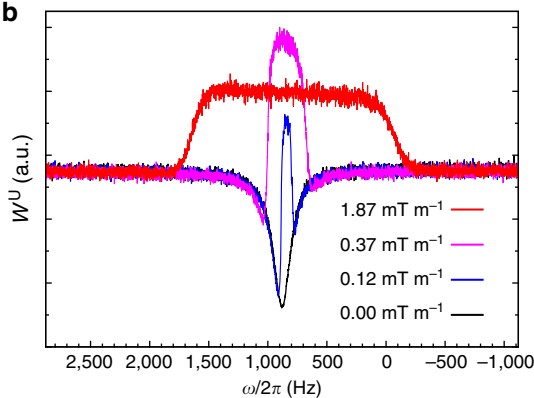

**Figure 1 | Influence of gradients on spin-noise spectra.** Simulated (**a**) and experimental (**b**) 700 MHz $^1$H spin-noise spectra acquired from a sample of 90% $H_2O$:10% $D_2O$ with a z-gradient of 0.00 (black), 0.12 (blue), 0.37 (magenta) and 1.87 mT m$^{-1}$ (red). The sample height used in the simulations was 20 mm, which corresponds approximately to that of the $^1$H saddle coil, while the true filling height in the round bottomed 5 mm o.d. sample tubes was 34 mm. The simulations corresponding to the experimental spectra were done using a Scilab programme[32] based on equation (5). The best-fit values extracted from the spectra of Fig. 3 were used for electronic parameters and radiation damping rate. A transverse relaxation rate of 2 Hz was assumed and the sample was split in 33 slices.

of less than perfect field homogeneity. These artifact peaks can be of advantage as indicators of insufficient field inhomogeneity as a necessary condition for their appearance is $\Delta\nu_{grad} \gtrsim \lambda_2$. In addition, they can provide the precise chemical shift of the major isotope peak in the radiation-damped spectra when determining isotope effects as described below.

**Secondary isotope effects**. To assess the typical situation of natural abundance $^{13}$C isotope-satellite peaks in $^1$H NMR, we performed simulations of proton NMR spectra in the presence of radiation damping. In Fig. 2 we show simulated spectra after a small-flip-angle excitation pulse (a), through a nuclear spin-noise detection scheme (b) to (d). Using equation (3) with parameters for a room-temperature probe (b) and a cryogenically cooled probe (c) for a spin system at room temperature, the response of a spin ensemble composed of a main resonance and two $^{13}$C satellites of 0.55% relative abundance, whose resonance frequencies are shifted by ±5 Hz relative to the main one. These parameters used are typical for the used experimental set-up (concentration, apparent $Q$ and $\eta$ values,...). The enhanced detection capabilities for isotopic effects achieved by resorting to a cryogenically cooled-coil probe and nuclear spin-noise detection

scheme can clearly be appreciated by the appearance of two narrow positive peaks stemming from the minor components superimposed on the broad main dip-shaped resonance (Fig. 2c). The enhancement allowed by cooled-coil probe set-up mainly results from the propensity of low-concentrated spin species to increase the spin-noise power level, while highly concentrated spins decrease the average noise power level[16]. As a consequence a significant 'bump on a dip' is observed for each satellite. This enhanced detection capability, not expected based on the fluctuation-dissipation theorem, in fact, results from the different temperatures of the noise sources. Indeed, if all sources are at the same temperature, the detection capability (Fig. 2b) is identical to the small-flip-angle excitation pulse (regime of linear response, Fig. 2a). The practical detection capability nevertheless strongly depends on the actual signal-to-noise ratio, the probe quality factor, sample concentration, the achievable spectral resolution and obviously the effect of the preamplifier noise (Supplementary Figs 2–5). For Fig. 2d, equation (5) was used with a set of parameters which corresponds to SNTO. Several features are noteworthy here. First, even if the chemical shift of the main line is set to $\omega_{main} = 0$ Hz, the resonance is shifted by frequency pushing, in agreement with observations recently reported[18]. This causes an improved detection capability for minor components when compared to Fig. 2c. Second, the separation of the two contributions (black curve for the coil and spin contributions, blue curve for the contribution due to the Nyquist noise of the preamplifier impedance) illustrates two features. First, that generally an in-phase Lorentzian line shape for the whole spectral density results from the summation of two phase-distorted Lorentzians[25]. Second, while the coil and spin contributions to the spectral density tend to enhance the detection capability of minor components (black curve), the contribution due to preamplifier impedance tends to reduce this capability since it decreases the noise level at the minor component resonance frequency. Obviously the final capability is strongly dependent on the $\mathscr{A}$, $\mathscr{B}$ and $\zeta$ values in equation (5), or in other words on the length of the transmission line. Experimentally and numerically, for a single species, the investigation of this dependence has revealed a large variety of situations[25].

In the experimental spin-noise spectra of the methyl protons of acetonitrile shown in Fig. 3, on top of the negative broad radiation-damped 1,2-$^{12}$C isotopomer signal, the primary and secondary $^{13}$C satellite doublets (due to coupling to 2-$^{13}$C and 1-$^{13}$C, respectively) appear as narrow positive peaks with intensities that differ from the natural isotopic abundance, due to the nonlinear feature of spin-noise response (Supplementary Figs 2–5). Selective decoupling of the nitrile $^{13}$C makes the secondary isotope peaks vanish, since their Larmor frequencies become too close to that of the main component: cross-precession effects become too efficient[28] (Supplementary Fig. 2). The primary and secondary isotope peaks appear shifted beyond any reasonable IECS from the broad main resonance since the latter is strongly affected by the frequency pushing effect, as discussed above and confirmed by the fitting curve (Supplementary Fig. 6). The difference of the coupled and 1-$^{13}$C decoupled spin-noise spectra shown in Fig. 3b is particularly suited to determine the small $^2J_{CH}$ and the IECS $^2\Delta$ $^1$H(C), as it is free from base line distortion. The true chemical shift (that is, devoid of frequency pushing) of the radiation-damped main peak can be determined from the small peak, which is due to an interference artefact seen in Fig. 3 resulting from static magnetic field inhomogeneity. From the analysis of Fig. 3, the following spectral characteristics are deduced $^2J_{CH} = 9.9 \pm 0.2$ Hz and $^2\Delta^1$H(C) = 1.1 ± 0.3 p.p.b. and $^1J_{CH} = 135.9 \pm 0.2$ Hz and $^1\Delta$ $^1$H(C) = 2.4 ± 0.3 p.p.b. Finally

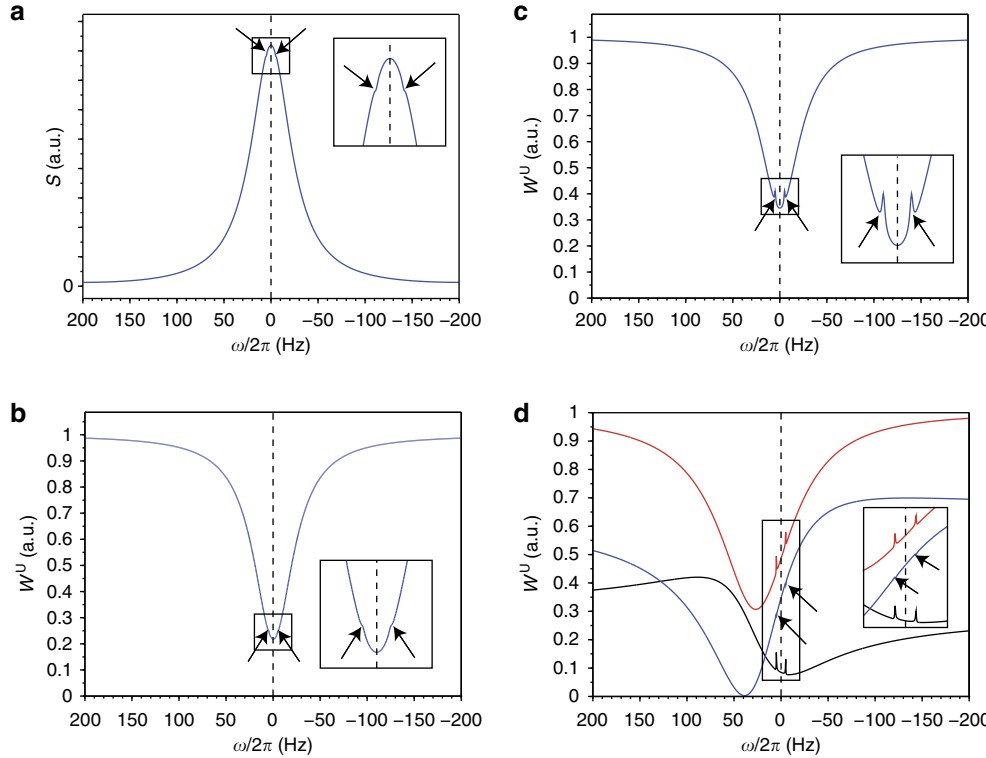

**Figure 2 | Simulations of isotope-satellite detection.** Simulations illustrate the improved detection capability of $^{13}C$ secondary isotopic effects (shown by arrows) allowed by resorting to cooled-coil probes and nuclear spin-noise detection scheme at spin-noise tuning optimum conditions. **a** corresponds to the signal response S after a small excitation pulse and Fourier transformation in the presence of radiation damping. **b** and **c** correspond to nuclear spin-noise spectra computed with equation (1) assuming a classical room-temperature probe ($\theta = 1$) and a cooled-coil ($\theta = 7.5$) probe, respectively. Finally, **d** illustrates the effect of the transmission line on nuclear spin-noise detection capability (equation (5)). The red curve corresponds to the observed spectral density (whole noise), the black curve to the contributions of the spin and coil noise, and the blue curve to the preamplifier noise.

from the best-fit theoretical curve (Supplementary Table 1), the phase $\arg(\zeta) \simeq 86.4°$ is found close to the $90°$ situation where radiation damping is fully quenched, since the current created by the precessing nuclear magnetization within the coil is dissipated in the preamplifier resistance. This feature is now exploited by spectrometer manufacturers for reducing radiation damping effects even if it can be at a price of a lower signal-to-noise ratio due to a smaller apparent quality factor $Q_{app}$ (ref. 15).

## Discussion

The nonlinear spin dynamics resulting from the interaction between sample magnetization and the detection coil can strongly alter the resonance line shape, providing a way to easily detect small NMR signals which are superimposed on large ones. This situation appears even more favourable when the detection is made through a nuclear spin-noise scheme using a probe with a cryogenically cooled coil. Our theoretical derivation is based on fluctuating rf fields originating from several sources (resistance of the coil and of the preamplifier, fluctuations of the transverse magnetization). From the analysis of the theoretical predictions, which are confirmed experimentally, two very specific spectral features are put forward. First and surprisingly a static magnetic field inhomogeneity induces the appearance of narrow spectral feature within the radiation-damped spin-noise resonance line shape, an effect enhanced when a cooled-coil probe is used. This discontinuity appears centered at the exact chemical shift, that is, it is not affected by a potential frequency pushing contribution from the resonant circuit. As a consequence it provides a robust solution to determine the absolute chemical shift of a resonance affected by radiation damping. Reciprocally one can determine

the frequency pushing contribution associated with the radiation damping feed-back field. Using the fluctuation-dissipation theorem, this is also true in small-flip-angle pulse experiments but without benefiting from the nonlinear amplification provided by noise sources at different temperatures.

A second application of the multi-spin species spin-noise spectroscopy consists in detecting small NMR signals obscured by a broad radiation-damped main resonance. As an illustration, we show that two-bond $^{13}C-^{1}H$ isotopic effects can be observed in $^{1}H$ nuclear spin-noise spectra recorded with cryogenically cooled probes on concentrated samples at natural isotope abundance. The main resonance appears as a negative signal, strongly broadened by radiation damping and the small satellite signals appear as narrow positive peaks. The nonlinear response enhancing their intensity allows their easy observation. Their spectroscopic features can be determined by a combination of decoupling difference spectroscopy and by fitting to the introduced equation (5).

More generally, the detection of minor components through nuclear spin noise is intrinsically connected to radiation damping rates, which can be enhanced by resorting to large concentrations, to tuning conditions other than the SNTO one such as the FSTO[18] (Supplementary Fig. 6 and Supplementary Table 1) or to probes with maximized $Q$ factors either through coil size reduction[30], or through dedicated cooled-coil probes with apparent $Q$ factor close to their effective value. As the effective length of the transmission line between the preamplifier and the detection coil affects the apparent $Q$ value and the noise contribution from the preamplifier, a great potential exists for optimization of radiation damping related effects and capabilities of the spin-noise scheme.

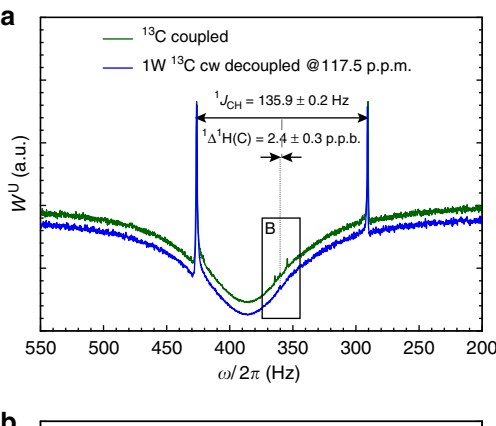

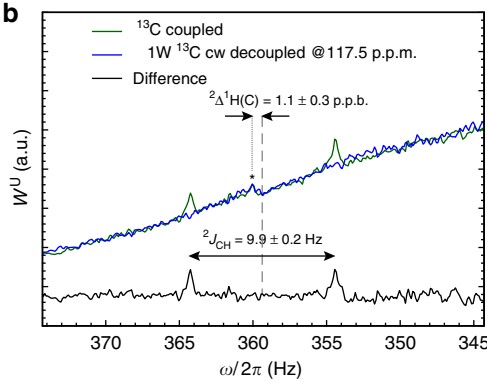

**Figure 3 | Experimental measurements of isotope satellites.** Experimental 700 MHz $^1$H spin-noise spectra at SNTO condition of 90% (v/v) acetonitrile with 10% (v/v) DMSO-$d_6$ for field frequency locking. (**a**) $CH_3$-region including the primary and secondary satellite doublets (green curve). The blue trace shows the selectively $^{13}$CN decoupled spectrum, slightly shifted downwards for better visibility. (**b**) The enlarged traces within the rectangle in **a** displayed together with their difference (black trace). The small peak between the satellites highlighted by an asterisk results from the interference between the residual magnetic field inhomogeneity and the radiation-damped main peak. It corresponds to the exact chemical shift of the $^{12}$C-methyl protons.

On a more general point of view, two directions of developments are expected to happen following the present work. First, the existence of an analytical solution, the observation of narrow spectral features induced by small magnetic field gradients and the improved understanding of the interactions between the detection circuit and the magnetization are roots on which improved protocols for classical high-resolution NMR, notably in biomolecular or metabolomic NMR can be developed. Most promising is the detection of resonances otherwise hidden by the radiation-damped solvent signals or for carefully determining the right frequencies at which optimal rf saturation of strong resonances can be achieved, that is, without frequency pushing contribution which is dependent on the longitudinal magnetization. Also, changes in the hardware design of the detection circuit are expected, as happened in the case of previous work on nuclear spin noise[15]. Second, the present study opens the route to non-invasive spectroscopic studies, lowering the detection limit for minor compounds, which are present in small ratios with respect to the main sample components. A very attractive feature of this scheme is the simplicity and the low-cost of the needed electronics. Typically, acquisition of nuclear spin-noise spectra only requires a suitable probe, a preamplifier and a spectrum analyzer. As a whole, this seems attractive for specific *in situ* chemical analysis exploiting NMR spectral features, in particular in the framework of a lab-on-chip[31].

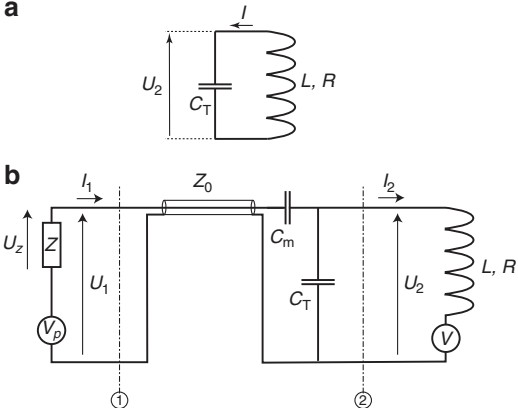

**Figure 4 | Electronic equivalent circuits considered.** (**a**) Simple model of the electronic circuit with a coil of inductance $L$ and resistance $R$ in parallel to a tuning capacitor $C_T$. The measurement is made at the capacitor extremities through the voltage $U_2$. (**b**) More realistic model of the used electronic circuit in NMR. The probe is constituted of a coil of inductance $L$ and resistance $R$ with a tuning capacitor $C_T$ in parallel and a matching capacitor $C_m$ in series. The probe is connected to the preamplifier of impedance $Z$ through a transmission line of impedance $Z_0$. The measurement is made at the $Z$ extremities through the voltage $U_z$. In this circuit several voltage sources are present. $V_p$ corresponds to the noise source due to the preamplifier resistance. $V$ corresponds to the voltage sources at the location of the coil, that is, the superposition of the noise due to the coil resistance, of the induction created by the magnetization fluctuations, $\mathscr{M}_s$, and of the induction created by the coherent precessing magnetization, $\mathscr{M}_r$. We denote $Z_{eq}$ the equivalent impedance of the left part of the electronic circuit at the plane number 2 ($Z$, transmission line and the two capacitors $C_T$ and $C_m$). The transmission matrix $T$ is between planes 1 and 2 and consequently contains the transmission line and the two capacitors, (in the derivation, it can contain other passive elements such as filters or other components present).

## Methods

**Theoretical derivation.** Here, only the main steps of the derivation of equation (5) are reported, the detailed derivation can be found in the Supplementary Methods. We first consider the electronic circuit depicted in Fig. 4a, the measurement is done at the extremities of the capacitor (voltage $U_2$). There are two physical processes which contribute to the observation of nuclear spin noise in NMR:[12]
- The fluctuations of the electric current in the coil induce the appearance of transient $B_1(t)$ magnetic fields which may be able to excite the nuclear magnetization, depending on their amplitudes and frequencies;
- The fluctuations of the transverse magnetization $\sum_m \mathscr{M}_s^m(t)$ create induced magnetic fields, $-i\mu_0 k(\Delta\omega_{LC})\sum_m \mathscr{M}_s^m(t)$, of a frequency almost equal to the spin Larmor resonance frequency, $\omega^m$.

In these conditions, denoting $\mu_0$ the magnetic permeability of free space, the time-dependent transverse magnetic field is:

$$B(t) = B_1(t) - i\mu_0 k(\Delta\omega_{LC})\sum_m \left[\mathscr{M}_s^m(t) + \mathscr{M}_r^m(t)\right] \quad (6)$$

where the third and last source of rf fields has been introduced. It corresponds to the feed-back field, $-i\mu_0 k(\Delta\omega_{LC})\sum_m \mathscr{M}_r^m(t)$, induced by the precessing coherent transverse magnetization, $\sum_m \mathscr{M}_r^m(t)$.

Since the magnetic field $B(t)$ is proportional to the current flowing in the coil, it can be related to the voltage $U_2(t)$:

$$U_2(t) = \alpha(\Delta\omega_{LC})B(t) \quad (7)$$

Working in the frequency domain, the Bloch equation for each spin species can be written as:

$$\mu_0 \mathscr{M}_r^m(\delta\omega^m) = 2\chi^m(\delta\omega^m)B(\delta\omega) \quad (8)$$

which illustrates the regime of linear response to small excitations.

The nuclear spin-noise spectrum corresponds to the spectral density of the voltage fluctuations:

$$W^U = \overline{U_2(\omega)U_2^*(\omega)} = |\alpha(\Delta\omega_{LC})|^2\overline{B(\omega)B^*(\omega)} \quad (9)$$

We then use the remark that $B_1(\delta\omega)$ fluctuations due to electric current in the coil are not correlated to those of the transverse magnetization, $\mathscr{M}_s^m(\delta\omega^m)$, and derived the relation between the spectral density of $\sum_m \mathscr{M}_s^m(\delta\omega)$ and that of $B_1(\delta\omega)$ using

an approach based on power flow[12] which is reminiscent from the demonstration of the fluctuation-dissipation theorem[10,11]:

$$\overline{\left|\sum_m \mathcal{M}_s(\delta\omega^m)\right|^2} = \frac{2\sum_m \chi^{m''}(\delta\omega^m)}{k'(\Delta\omega_{LC})}|B_1(\delta\omega)|^2 \qquad (10)$$

Combining the different equations, and applying modifications to take into account the potential difference of temperatures between the coil and the sample (coefficient $\theta$) and the case of hyperpolarized species (coefficients $K_m$), we finally obtain equation (1). At $\Delta\omega_{LC} = 0$, the nuclear spin-noise spectrum corresponds to an in-phase Lorentzian line shape for non-overlapping nuclear spin resonances (equation (3)), as described by McCoy and Ernst[13] for a single-spin species.

The extension to the electronic circuit of Fig. 4b requires to take into account the dephasing effect of the transmission line and the finite value of the preamplifier impedance $Z$. There are now three sources of rf-field fluctuations: the two previous ones (due to the transverse magnetization and Nyquist noise of the coil, which contribute to the random source of voltage $V$ represented in Fig. 4b) and a third one, due to the Nyquist noise of the $Z$ impedance, which contributes to the voltage source $V_p$. Since there is no correlation between the different noise sources, the principle of superposition can be used, with the final voltage $U_z = U_z^c + U_z^p$ where $U_z^c$ is the measured voltage due to fluctuations occurring in the coil (magnetization and coil resistance) and $U_z^p$ to that due to the preamplifier impedance.

Using the transmission matrix $T = \begin{pmatrix} T_{11} & T_{12} \\ T_{21} & T_{22} \end{pmatrix}$ between planes 1 and 2 and the equivalent impedance $Z_{eq}$ of the left part of the circuit seen at plane 2 of Fig. 4b, the measured voltage $U_z^c$ is found proportional to $V$:

$$U_z^c = \frac{Z_{eq}T_{22} + T_{12}}{Z_{eq} + jL\omega + R}V \qquad (11)$$

As a consequence, the expression of coefficient $\mathcal{A}$ of equation (1) has to be modified for considering the whole electronic circuit described in Fig. 4b but the frequency dependence remains identical. The two sources of fluctuations (transverse magnetization and Nyquist noise of the coil) induce a contribution to the nuclear spin-noise spectral density given by the first term of equation (5).

For the contribution due to the preamplifier noise source, there are two voltage sources $V_p$ due to Nyquist noise and $V$ due to the precessing magnetization ($\propto \sum_m \mathcal{M}_r^m$). Still using the transmission matrix, the measured voltage $U_z^p$ is found to be a linear combination of $V_p$ and $V$. $\sum_m \mathcal{M}_r^m$ is found to be proportional to rf excitations $B_1^p(t)$ through the nuclear susceptibility (equation (8)). Thanks to the use of the transmission matrix $T$, voltage fluctuations $V_p$ are also shown to be proportional to rf excitations $B_1^p(t)$. Combining these relations, the preamplifier noise source is found to contribute to the noise spectral density through the second term of equation (5), the one proportional to $\mathcal{B}$ (See Supplementary Methods).

**Experimental.** NMR experiments were run on Bruker spectrometers at room temperature. Spectra at 700 MHz $^1$H frequency on an Avance III spectrometer equipped with a 5 mm TCI probe with an rf-coil cooled to 20 K. All spin-noise spectra were acquired as pseudo two-dimensional series of long (15.6 s for Fig. 1, 41.5 s for Fig. 3) noise blocks (for an overall acquisition of 0.5 and 6 h, respectively). During the processing, each block was split according to the chosen final resolution (1.0 Hz for Fig. 1 and 0.2 Hz for Fig. 3) after zero-filling (none for Fig. 1, by a factor 2 for Fig. 3) and the sliding windows principle was used to enhance the signal-to-noise ratio[6,21]. The $^{13}$C decoupled spin-noise spectra in Fig. 3 were obtained by low power (1.6 kHz) continuous wave rf irradiation at the $^{13}$CN Larmor frequency (117.5 p.p.m.).

All numerical simulations were performed using SciLab[32]. For Fig. 2, we chose parameters similar to the experimental ones: for the three species $\lambda_2 = 4$ Hz, concentration of the main component 40 mol l$^{-1}$, same chemical shift. For Fig. 2 a, b and c, $Q_{app} = 700$, $\eta = 0.07$, $B_0 = 16.4$ T, $W_a^U$: 20% of the coil noise level, resolution 0.25 Hz. $Q_{app}$ is typically reduced by a factor 10 when compared with the coil quality factor ($L\omega_0/R$) by manufacturers' choices of transmission line length. For Fig. 2d and Fig. 1, the used electronic parameters $(\mathcal{A}, \mathcal{B}, k(\Delta\omega_{LC}), \theta, \zeta, W_a^U)$ were those deduced from the best-fit theoretical curve of Fig. 3. For that purpose, a specific programme based on the Levenberg-Marquardt algorithm was used[33]. The best-fit values were: $\mathcal{A} = 0.3$, $\mathcal{B} = 0.65$, $W_a^U = 0.05$, $\theta = 15.5$, $|\zeta| = 0.74$, $\arg(\zeta) = -86°$, $\arg(k(\Delta\omega_{LC})) = -29°$, $\lambda_2 = 1.3$ Hz and $\lambda_r = 324$ Hz for the $^{12}$C-methyl resonance. Due to the large line width of the main peak caused by the significant radiation damping contribution, its resonance frequency determination is, in fact, limited[34].

**Data availability.** The data that support the findings of this study are available from the corresponding author upon request.

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

## Acknowledgements

This research was supported by the Austrian Science Funds (FWF) and the Agence Nationale de Recherche (ANR) in the joint project IMAGINE (FWF project number I1115-N19, ANR project 12-IS04-0006) and by the EGIDE-ÖAD AMADEUS Austrian-French exchange programme (no 28948WD and FR 11/2013). The Avance III 700 MHz NMR spectrometer was co-financed by the European Union through the INTERREG IV ETC-AT-CZ programme (EFRE RU2-EU-124/100-2010, project M00146 'RERI-uasb'). Bérangère Dubrulle-Bréon is acknowledged for helpful discussions on fluctuation-dissipation theorem.

## Author contributions

N.M. and H.D. designed and directed the project; M.T.P., V.V.R., J.S., N.M. and H.D. performed the experiments; M.T.P. and H.D. analysed spectra; N.M. and H.D. made the simulations; H.D. developed the theoretical framework; M.T.P., N.M. and H.D. wrote the article.

## Additional information

**Competing financial interests:** The authors declare no competing financial interests.

**Publisher's note**: 

