## [Peer Review File · Nature Communications]

Reviewers' comments:

Reviewer #1 (Remarks to the Author):

The manuscript by Desvaux and colleagues describes a novel application of spin-noise spectroscopy, a concept developed by Slean, Hahn and coworkers and by McCoy and Ernst many years ago but which has remained a curiosity till today. In this manuscript, small signals that are typically hidden behind large, radiation-damped signals are detected in a nonlinear process where signal from the strong resonance also acts back on the species giving rise to the small signal.

This is an excellent piece of work. However, the complex formalism required and the fact that the complex formula can only be evaluated numerically, makes the manuscript extremely difficult to understand except for a small circle of experts. The fact that most of the derivation is relegated to the SI information does not help in this respect. Equation 1 (referred to as equation 10 in the text) is not familiar to a typical reader and needs to be introduced more carefully.

The statement "the present study opens the route to spectroscopy through nuclear spin noise. This is illustrated by the detection and the characterization of small secondary isotopic effects on ^1H due to natural low abundance content of ^{13}C isotopes." is hard to substantiate from the results presented. To make the method attractive to a wide range of spectroscopists, application to a protein or similar would be necessary, in my view. I am convinced that the information extracted from acetonitrile could also be obtained by the classical methods referenced in the introduction.

I therefore recommend that the full formalism is included into the main text and that the paper is published in a more specialized journal, e.g. the Journal of Chemical Physics.

.

Minor points.

- I find the introduction not to the point. It should focus on the topic. Links to environmental chemistry, for example, are distracting and unnecessary.
- Figure 2: the important features are almost invisible and should be enlarged in an inset.
- All figures: Axis labels and marks are too small to read; the grids probably reproduce badly in the journal.

Reviewer #2 (Remarks to the Author):

The paper describes a novel technique for measuring nuclear magnetic resonance (NMR) lines using spin noise detection. With normal NMR, an rf field is generated by the spectrometer to excite the spin system and the response is detected, often in the form of a free induction decay of the transverse magnetization or a spin echo. In this work there is no excitation from the spectrometer but noise sources generate excitations of the spin system through coupling of the spins to the detection coil. There are 2 advantages to this scheme over pulsed NMR. If the magnetization is high there can be strong radiation damping effects that distort the signals: these effects are absent using spin noise. Secondly, it is normally very hard to select signals arising from coupling to low abundance isotopes in the presence of the main NMR line. This appears to be possible using the methods described in the paper.

I have no doubt about the novelty, timeliness and significance of the work.

The paper is well written and prepared, apart from a few grammatical corrections that the editors can find.

I do however feel that the abstract could be improved. I am aware that Nature Comms is for specialists but to appeal to a slightly wider audience I believe the abstract could provide a more

accessible introduction. e.g. define an NMR set-up with detection coil feeding a receiver; define NMR; etc. Just a couple of sentences would help boost the audience. Also I suggest moving the key findings (e.g. last 2 sentences) towards the start.

In summary, with some attention to the abstract, I believe the article is suitable for publication in NC.

Reviewer #3 (Remarks to the Author):

This work reports an intriguing application of NMR without coherent rf excitation. The noise fluctuations of the NMR set-up at high field with a high-quality detection circuit are employed to detect the NMR signal. The authors identify three noise sources that need to be considered in their theory which nicely describes the observed phenomena. Apart from advancing the understanding of intricacies of the selection electronics in NMR spectrometers, the authors show the use of their spin-noise detection scheme to quantify the chemical shift variation due to the presence of rare isotopes one and two bonds away from the observed protons. This shift is in the ppb range and hard to observe.

This manuscript is fascinating to the NMR expert. It is doubtful, however, in how far it can catch the interest of a more general readership. For this I am missing reference to the famous fluctuation-dissipation theorem and a brief explanation of the steps that NMR takes us beyond this stage. Moreover, the advantage of spin-noise detection over other forms of detecting secondary isotopic chemical shifts need to be stated clearly. In its current state, the reader gets the impression, that the manuscript serves to sell spin-noise NMR as a useful NMR technique by example of mastering the difficulties of detecting the isotope effect. The broader benefits of spin-noise NMR do not become clear, and the reader is left puzzled with the question of what the larger implications of this study are. The manuscript would certainly be well placed in a hard.core NMR journal.

Reviewer #4 (Remarks to the Author):

This is a very nice paper by a group which has focused on nonlinear magnetic resonance dynamics for a very extended period. It has some significant extensions of the theoretical treatment of lineshapes under spin noise, and a good discussion of the underlying physics.

However, I have no idea why they would have submitted this to Nature Communications, as the work is essentially of no interest to people outside of a very specialized physics community. The "observation" they use to validate the technique (two-bond isotopic changes to the chemical shift) are, in their work, on order 1 Hertz (ca. 1 ppb in a 700 MHz NMR), and could be instantly observed simply by taking the NMR spectrum of the dilute molecule. Further, it is very unlikely this can be generalized to more complex species, with many peaks, as it is certain the competitive effects of multiple strong peaks would be highly confounding. So there is no possibility this will lead to a useful technique for any of the applications discussed at the beginning of the paper.

If this paper were submitted to Nature Physics, I would have supported acceptance (and if you want to refer it, you can quote me) because the physics is elegant. It is just that it absolute is not of general interest.

We would like to acknowledge the reviewers for their works and comments which were useful inputs for improving our manuscript. We were particularly delighted by their recognition of the originality of our work. Before addressing specifically each comment we shall start with a general one which questioned the choice of *Nature Communications* for submitting this work.

The first point to mention was our choice not to split our work in several articles. In fact, there are three significant results, but we found much more convincing to report them simultaneously.

- The first main result is the derivation of the analytical solution for the nuclear spin-noise spectral density of a system with several spin species, also considering the magnetization excitation due to the preamplifier impedance (which was previously unsolved analytically even for a single-spin species). It represents a significant step forwards since it allows predictive simulations. This part is typically a physical or a chemical-physics result.
- The second result concerns the observation of static magnetic-field gradient effects on spin-noise spectra. This represents an attractive aspect, for instance, for qualifying magnetic field homogeneity in NMR, or obtaining absolute chemical shift in the presence of radiation damping. It typically corresponds to instrumental developments with very short application time, and would be suitable for a magnetic resonance journal.
- The third result is more chemically oriented since it concerns the detection of minor components by nuclear spin noise and notably, the capability of measuring isotopic effects. This aspect concerns analytical chemistry. It is expected to have a longer application time than the second point but a potential larger scale use. Indeed if the demonstration is done, the usefulness of the method would depend on the considered specific applications, and would require specific instrumental developments for taking benefit of the whole non-linear features of the approach and the reduced request in terms of instrumentation when compared to classical NMR. It remains that in analytical chemistry isotopic measurements have large and attractive applications with some very success story even if it was not clear at the beginning. A nice illustration is the introduction of the SNIF-NMR method in 1982 which leads today to the world-wide company EuroFINS. That also explains why we have voluntarily emphasized our introduction on the potential applications of isotopic effects, even if our approach concerns the detection of minor components.

As a consequence, with a work in physics which has impact, expected to be large, in analytical chemistry and instrumentation, we decided to submit it to a multi-disciplinary journal (chemistry and physics) which accepts manuscripts, length of which is compatible with an article. The choice of *Nature Communications* then appears rather natural. Since this journal accepts pure physics articles but also multi-disciplinary ones. Moreover this choice was strengthened by advertisements from this journal for accepting articles in physical chemistry and chemical physics.

Important changes were made in the manuscript for addressing reviewers' comments but also to agree with the journal editorial recommendations. They, in particular, concern the abstract, the introduction and the Supporting Information. All significant changes are highlighted by text written in blue.

Reviewers' comments:

Reviewer #1 (Remarks to the Author):

The manuscript by Desvaux and colleagues describes a novel application of spin-noise spectroscopy, a concept developed by Slean, Hahn and coworkers and by McCoy and Ernst many years ago but which has remained a curiosity till today. In this manuscript, small signals that are typically hidden behind large, radiation-damped signals are detected in a nonlinear process where signal from the strong resonance also acts back on the species giving rise to the small signal.

This is an excellent piece of work. However, the complex formalism required and the fact that the complex formula can only be evaluated numerically, makes the manuscript extremely difficult to understand except for a small circle of experts.

We, only partially agree with this comment. We, in fact, introduced an analytical solution. Effectively its expression is not simple, but it has to be compared to the equation describing overlapping lines in the frequency domain with classical NMR; not simple too... If the lines do not overlap, the power spectrum can be described by a sum of Lorentzian-type functions, one for each resonance.

The fact that most of the derivation is relegated to the SI information does not help in this respect. Equation 1 (referred to as equation 10 in the text) is not familiar to a typical reader and needs to be introduced more carefully.

In the spirit of *Nature Communications*, the text (results and Methods) should be kept short. We, nevertheless, prefer to report the whole demonstration in SI for the completeness. Nevertheless following the reviewer's comments we have completed the theoretical description in the Results section, ordered the equations, and provided the main equations in the Results section.

The statement "the present study opens the route to spectroscopy through nuclear spin noise. This is illustrated by the detection and the characterization of small secondary isotopic effects on ^1H due to natural low abundance content of ^{13}C isotopes." is hard to substantiate from the results presented. To make the method attractive to a wide range of spectroscopists, application to a protein or similar would be necessary, in my view. I am convinced that the information extracted from acetonitrile could also be obtained by the classical methods referenced in the introduction.

Extension to protein NMR is well beyond the present development and we do not believe that spin-noise approach will be directly used for this type of applications in the next future. But, protein NMR can benefit from the current finding, by improving the probe sensitivity (Cf. Ref. [17]) or by being able to qualify the probe homogeneity. Moreover NMR is not only dedicated to biomolecules, it can be used for chemical analysis and in this field for specific purposes, spin-noise detection can represent an attractive solution. The end of the discussion was modified to emphasize this point.

I therefore recommend that the full formalism is included into the main text and that the paper is published in a more specialized journal, e.g. the *Journal of Chemical Physics*.

This point has been addressed in the introduction of the present letter.

. Minor points.

- I find the introduction not to the point. It should focus on the topic. Links to environmental chemistry, for example, are distracting and unnecessary.

The introduction has been largely rewritten but we kept analytical chemistry examples from several fields such as environmental since isotopic measurements are really important in this field. Moreover we also enlarged the potential application field considering detection of minor components.

- Figure2: the important features are almost invisible and should be enlarged in an inset.

The important features have been enlarged; it is also the case in others figures.

- All figures: Axis labels and marks are too small to read; the grids probably reproduce badly in the journal.

All figures have been modified according to the reviewer's comment.

Reviewer #2 (Remarks to the Author):

The paper describes a novel technique for measuring nuclear magnetic resonance (NMR) lines using spin noise detection. With normal NMR, an rf field is generated by the spectrometer to excite the spin system and the response is detected, often in the form of a free induction decay of the transverse magnetization or a spin echo. In this work there is no excitation from the spectrometer but noise sources generate excitations of the spin system through coupling of the spins to the detection coil. There are 2 advantages to this scheme over pulsed NMR. If the magnetization is high there can be strong radiation damping effects that distort the signals: these effects are absent using spin noise. Secondly, it is normally very hard to select signals arising from coupling to low abundance isotopes in the presence of the main NMR line. This appears to be possible using the methods described in the paper.

I have no doubt about the novelty, timeliness and significance of the work.

The paper is well written and prepared, apart from a few grammatical corrections that the editors can find.

I do however feel that the abstract could be improved. I am aware that Nature Comms is for specialists but to appeal to a slightly wider audience I believe the abstract could provide a more accessible introduction. e.g. define an NMR set-up with detection coil feeding a receiver; define NMR; etc. Just a couple of sentences would help boost the audience. Also I suggest moving the key findings (e.g. last 2 sentences) towards the start.

We have fully modified the abstract also because our previous version did not agree with *Nature Communications* requests. We also mention in the new version, as suggested by the reviewer the importance in the change of the needed electronics.

In summary, with some attention to the abstract, I believe the article is suitable for publication in NC.

Reviewer #3 (Remarks to the Author):

This work reports an intriguing application of NMR without coherent rf excitation. The noise fluctuations of the NMR set-up at high field with a high-quality detection circuit are employed to detect the NMR signal. The authors identify three noise sources that need to be considered in their theory which nicely describes the observed phenomena. Apart from advancing the understanding of intricacies of the selection electronics in NMR spectrometers, the authors show the use of their spin-noise detection scheme to quantify the chemical shift variation due to the presence of rare isotopes one and two bonds away from the observed protons. This shift is in the ppb range and hard to observe.

This manuscript is fascinating to the NMR expert. It is doubtful, however, in how far it can catch the interest of a more general readership. For this I am missing reference to the famous fluctuation-dissipation theorem and a brief explanation of the steps that NMR takes us beyond this stage.

We acknowledge the reviewer for his comment which allows us to enlarge some conclusions, notably in the capability of detecting small magnetic field inhomogeneity through classical detection scheme using small-flip angle pulse (but not as straightforward as with nuclear spin noise with a cooled-coil probe). We have introduced the fluctuation-dissipation theorem in the Introduction, in the Results and in the Methods sections and its link with the present works.

Moreover, the advantage of spin-noise detection over other forms of detecting secondary isotopic chemical shifts need to be stated clearly. In its current state, the reader gets the impression, that the manuscript serves to sell spin-noise NMR as a useful NMR technique by example of mastering the difficulties of detecting the isotope effect. The broader benefits of spin-noise NMR do not become clear, and the reader is left puzzled with the question of what the larger implications of this study are. The manuscript would certainly be well placed in a hard.core NMR journal.

We have changed the end of discussion to address the comment on the potentiality of the approach.

Reviewer #4 (Remarks to the Author):

This is a very nice paper by a group which has focused on nonlinear magnetic resonance dynamics for a very extended period. It has some significant extensions of the theoretical treatment of lineshapes under spin noise, and a good discussion of the underlying physics.

However, I have no idea why they would have submitted this to Nature Communications, as the work is essentially of no interest to people outside of a very specialized physics community. The "observation" they use to validate the technique (two-bond isotopic changes to the chemical shift) are, in their work, on order 1 Hertz (ca. 1 ppb in a 700 MHz NMR), and could be instantly observed simply by taking the NMR spectrum of the dilute molecule. Further, it is very unlikely this can be generalized to more complex species, with many peaks, as it is certain the competitive effects of multiple strong peaks would be highly confounding. So there is no possibility this will lead to a useful technique for any of the applications discussed at the beginning of the paper.

Since chemical shift is concentration dependent, diluting the sample cannot be said as a perfect response. This aspect is now addressed in the Introduction. Moreover the end of the Discussion section has been changed to address this point. In particular, we think that a space exists for spin-noise detection for continuous monitoring with a lab-on-chip design.

If this paper were submitted to Nature Physics, I would have supported acceptance (and if you want to refer it, you can quote me) because the physics is elegant. It is just that it absolute is not of general interest.

The point on the choice of the journal has been addressed in the first part of this letter.

REVIEWERS' COMMENTS:

Reviewer #2 (Remarks to the Author):

I was in favor of publication provided the abstract was improved. This the authors have done. Probably the last sentence in the abstract should read two-bond without s.

As to whether this is the appropriate journal most likely depends on one's view of what constitutes general interest. I did have concerns about this aspect of the article which is why I suggested improvements to the abstract to draw in non-NMR experts if possible. I think it is marginal but I am inclined to the believe that it suitable for NC.

Response to the reviewers's comments.

Reviewer #2 (Remarks to the Author):

I was in favor of publication provided the abstract was improved. This the authors have done. Probably the last sentence in the abstract should read two-bond without s.

This correction has been done.

As to whether this is the appropriate journal most likely depends on one's view of what constitutes general interest. I did have concerns about this aspect of the article which is why I suggested improvements to the abstract to draw in non-NMR experts if possible. I think it is marginal but I am inclined to the believe that it suitable for NC.